# Intensive Care Outcomes of Patients after High Dose Chemotherapy and Subsequent Autologous Stem Cell Transplantation: A Retrospective, Single Centre Analysis

**DOI:** 10.3390/cancers12061678

**Published:** 2020-06-24

**Authors:** Panagiotis Karagiannis, Lena Sänger, Winfried Alsdorf, Katja Weisel, Walter Fiedler, Stefan Kluge, Dominic Wichmann, Carsten Bokemeyer, Valentin Fuhrmann

**Affiliations:** 1Department of Oncology, Haematology and Bone Marrow Transplantation with Section of Pneumology, University Medical Centre Hamburg-Eppendorf, Martinistrasse 52, 20246 Hamburg, Germany; w.alsdorf@uke.de (W.A.); k.weisel@uke.de (K.W.); fiedler@uke.de (W.F.); cbokemeyer@uke.de (C.B.); 2Department of Intensive Care Medicine, University Medical Centre Hamburg-Eppendorf, 20246 Hamburg, Germany; lenahumbracht@outlook.de (L.S.); s.kluge@uke.de (S.K.); d.wichmann@uke.de (D.W.); vfuhrmann@outlook.de (V.F.)

**Keywords:** high dose chemotherapy, autologous stem cell transplantation, intensive care unit

## Abstract

High dose chemotherapy (HDT) followed by autologous peripheral blood stem cell transplantation (ASCT) is standard of care including a curative treatment option for several cancers. While much is known about the management of patients with allogenic SCT at the intensive care unit (ICU), data regarding incidence, clinical impact, and outcome of critical illness following ASCT are less reported. This study included 256 patients with different cancer entities. Median age was 56 years (interquartile ranges (IQR): 45–64), and 67% were male. One-year survival was 89%; 15 patients (6%) required treatment at the ICU following HDT. The main reason for ICU admission was septic shock (80%) with the predominant focus being the respiratory tract (53%). Three patients died, twelve recovered, and six (40%) were alive at one-year, resulting in an immediate treatment-related mortality of 1.2%. Independent risk factors for ICU admission were age (odds ratio (OR) 1.05; 95% confidence interval (CI) 1.00–1.09; *p* = 0.043), duration of aplasia (OR: 1.37; CI: 1.07–1.75; *p* = 0.013), and Charlson comorbidity score (OR: 1.64; CI: 1.20–2.23; *p* = 0.002). HDT followed by ASCT performed at an experienced centre is generally associated with a low risk for treatment related mortality. ICU treatment is warranted mainly due to infectious complications and has a strong positive impact on intermediate-term survival.

## 1. Introduction

The number of malignant diseases is increasing world-wide and will reach over 22.2 Mio/years by the end of 2020. Approximately 13.5–21.5% of patients diagnosed with cancer are admitted to intensive care units (ICUs) during the course of their disease [1,2]. In the past, a majority of mostly solid cancer patients but also haematological cancer patients were denied the admission to the ICU because of an anticipated poor prognosis for their cancer. This was due to the fact that the underlying medical condition was not treatable and would lead to death during the next months. Only recently, a growing number of studies have proven that critically ill patients with cancer can benefit from the treatment at the ICU [3]. In the last decade, survival rates of patients with cancer have increased to around 50–60%. Several authors suggested that full supportive care should be provided to patients with a life expectancy of at least 1 year as an arbitrary cut-off point [4].

High dose chemotherapy (HDT) with subsequent autologous peripheral blood stem cell transplantation (ASCT) is currently a standard treatment option in different cancers [5]. Remarkably, HDT has the potential to cure relapsed aggressive lymphomas or relapsed and refractory germ cell tumours [6,7]. Conditioning regimens lead to significant myelosuppression. Doses are maximized in order to achieve higher concentrations with higher tissue penetration and to potentially overcome resistance to conventionally dosed chemotherapy. However, neutropenia is expected to be severe, and patients often need red blood cell and platelet transfusion as well as intravenous antibiotic treatment. Overall, the predominant complications of HDT are infections [8].

Autologous stem cell transfusion has been introduced to shorten the time of aplasia after HDT to mostly 5–10 days [9,10]. With the adoption of peripheral blood stem cells rather than bone marrow as stem cell source and improvements in supportive care, mortality after HDT and ASCT has been markedly reduced over the last decades [11].

Although HDT has become part of the established treatment algorithm in serval malignancies, the possibility of severe complications remains. Data concerning the need and the frequency of ICU admission following high dose chemotherapy and subsequent autologous stem cell transplantation are scarce [12,13]. Therefore, we studied the clinical course of patients with different cancer entities receiving high dose chemotherapy and subsequent ASCT focusing on their outcome and performance on the ICU in a specialised referral centre.

## 2. Results

In total, 256 patients who received high dose chemotherapy with subsequent autologous stem cell transplantation in a tertiary medical centre were included in this analysis. Median age of the patients was 56 years (interquartile ranges (IQR): 45–64), and 67% (*n =* 172) were male. One hundred and four patients were older than sixty years of age. Patient distribution was as follows: 19% (*n* = 48) had a germ cell tumour, 48% (*n* = 122) a multiple myeloma, 19% (*n* = 49) non-Hodgkin’s lymphoma (NHL), 7% (*n* = 18) central nervous system-lymphoma (CNS-L), 3% (*n* = 8) Hodgkin’s lymphoma (HL), and 4% (*n* = 11) sarcoma and miscellaneous tumours (Table 1). 

The main high dose chemotherapy protocols included melphalan (*n* = 125), high dose carmustine/etoposide/cytarabine/melphalan (BEAM; *n* = 54), carboplatin/etoposide (CE; *n* = 41), cisplatin/etoposide/ifosfamide (HD-PEI; *n* = 10), carmustine/thiotepa (*n* = 18) and several other chemotherapies (*n* = 8) (Appendix A).

Treatment-related mortality during HDT and subsequent ASCT for all patients was 1.2%. The subgroup of sarcoma and miscellaneous rare occurring tumours had the worst prognosis with a median survival of 167 days. For all other entities, median survival was not reached during the observation period (Figure 1).

Fifteen patients (cohort 1) had to be admitted to the ICU for a complication during high dose chemotherapy treatment (HDT) and subsequent ASCT. This group included 13% of patients suffering from germ cell tumour, 20% multiple myeloma, 33% NHL, 13% CNS-L, and 20% miscellaneous tumours (Table 1).

The median age of the patients was 64 (IQR 43–67) years. The main reason for ICU-admission was septic shock (80%) with the predominant focus of infection being the respiratory tract (53%). Other reasons included ventricular tachycardia, seizure, and Ileus (Table 2). 

Laboratory parameters in cohort 1 such as leucocyte count and c-reactive protein (CRP) did not correlate with disease outcome (Appendix A). Twelve patients (80%) survived their stay at the ICU, and six (40%) were alive at one-year. Independent risk factors for ICU admission following HDT and ASCT were age at HDT (odds ratio (OR) 1.05; 95% confidence interval (CI) 1.00–1.09; *p* = 0.043), duration of aplasia (OR: 1.37; CI: 1.20–2.23; *p* = 0.013), and the Charlson comorbidity index (OR: 1.64; CI: 1.20–2.23; *p* = 0.002) (Table 3A). In contrast, parameters such as haemoglobin, thrombocyte count, quick and INR did not reach significance in predicting the ICU admission. Furthermore, using a calculated cut-off point for aplasia (6.5 days) and Charlson comorbidity sum (3.5), odds ratios were 4.84 (CI: 1.61–14.45; *p* = 0.005) and 5.65 (CI: 1.71–18.62; *p* = 0.004), respectively (Table 3B).

Thirty-one patients (cohort 2) had to be admitted to the ICU after successfully completing HDT and ASCT within 6 months of treatment. Of these patients, 16 (52%) were germ cell tumour, five (16%) multiple myeloma, six (19%) non-NHL, one (3%) CNS-L, one (3%) HL, and two (6%) miscellaneous tumours (Table 1). Cohort 2 had a median age of 44 years, and the main reason for admittance was elective surgery (*n* = 19) due to tumour resection. Further reasons were sepsis (*n* = 4), respiratory insufficiency (*n* = 3), bleeding (*n* = 3), and neurological impairment (*n* = 2) (Table 2). 

The sequential organ failure score (SOFA) was significantly higher in cohort 1 versus cohort 2 (8 (IQR 6–12) versus 4 (IQR 2–8), *p* = 0.001). These findings correlated with more supportive medical aid in cohort 1 such as invasive ventilation (53% versus 6%), the need of vasopressors (67% versus 11%), and renal replacement therapy (33% versus 6%) (Table 2). Furthermore, patients in cohort 1 had a median ICU stay of 8 (IQR 2–27) days compared to patients in cohort 2, who had a median stay of 1 (IQR 1–5) day; total hospital stay was 46 (IQR 24–75) versus 14 (IQR 10–27) days, respectively. In cohort 1, 47% of patients had a normal ejection fraction (EF > 55%) compared to 45% in cohort 2. Infectious complications were less frequent in cohort 2 (26%) compared to cohort 1 (80%) during the ICU stay. In cohort 1, 53% were diagnosed with a hospital acquired pneumonia compared to 19% in cohort 2. Cohort 1 had one case of herpes simplex virus (HSV) pneumonia and five (33%) cases of severe fungal infections including one case of candidemia compared to cohort 2, which had four (13%) cases of severe fungal infection and no candidemia. Cohort 1 had nine detected blood stream infections predominantly with coagulase negative staphylococcus compared to only two in cohort 2 (Table 4).

A detailed microbiological analysis is found in Table 5. The 365-day mortality after ICU discharge in cohort 1 was 50% compared to 27% in cohort 2, of which 33% was due to tumour progression in cohort 1 compared to 17% in cohort 2.

## 3. Discussion

To the best of our knowledge, this is the first comprehensive study comparing the safety of HDT and ASCT across different cancer entities including solid tumours. Furthermore, the study analysed the clinical outcome of two groups of these patients treated in the ICU.

In the study cohort of 256 patients, ASCT performed in an experienced centre was safe and had a low risk for treatment related mortality, as seen by an immediate treatment-related mortality of 1.2% for those patients. No treatment groups apart from the subgroup of sarcoma and rare cancer entities reached their median survival in the study period of up to 8 years. Moreover, the survival rate of multiple myeloma, the largest cancer entity in this analysis, was comparable with reported data indicating that this cohort was a representative sample for this entity [14].

Nevertheless, sarcoma and rare cancer entities had the worst survival, reaching a median survival after only 167 days. This finding reflects that curative treatment options for sarcoma are still not satisfactory [15,16].

Six percent of our study population required treatment in immediate relation to their high dose chemotherapy at the ICU, and 80% of these patients were discharged after full life support. After one year, 50% of the patients discharged from ICU were alive, indicating prolonged survival after ICU therapy [17,18]. A recent consensus statement outlines the requirements for admission of cancer patients to the ICU. The authors of the consensus statement defined recommendations for invasive measures such as mechanic ventilation and the use of dialysis, concluding a clear benefit of the ICU treatment [19]. Patients that were admitted to the ICU in relation to their HDT and ASCT treatment (cohort 1) had a statistically significant higher SOFA score compared to patients being admitted within 6 months after HDT and ASCT (*p* = 0.001), showing that scoring systems in ICU are also reliable tools in oncological patients [20]. The higher SOFA score correlated with more supportive medical aid such as invasive ventilation (67% versus 35%), the need for vasopressors (53% versus 6%), and dialysis (33% versus 6%). Moreover, patients within cohort 1 had not only a statistically significant longer ICU stay (*p* = 0.0004) but also prolonged hospital stays (*p* < 0.0001). Importantly, only 26% of patients admitted to the ICU in immediate relation to the treatment (cohort 1) died due to progression in tumour disease within 365 days. This indicates that the decision for ICU admission should only be made due to the prevailing medical condition and should not be influenced by the severity of their cancer diagnosis or the arbitrary 1 year survival rate in this group of patients [21].

Furthermore, microbiological analysis showed that cohort 1 had more gram-positive blood stream infections compared to cohort 2 (nine versus three). Most of these infections were triggered by coagulase negative staphylococcus. Patients in cohort 1 had more antibiotic resistant bacteria detected in their total hospital stay (four vancomycin-resistant enterococcus, two extended spectrum β-lactamase producing bacteria) compared to cohort 2 (one vancomycin-resistant enterococcus and one extended spectrum β-lactamase producing bacteria), which required an extended antibiotic treatment regime. This high prevalence of resistant organisms might be associated with a more severe course of disease [22,23]. Interestingly, patients who had been admitted to the ICU within 6 months of successful treatment (cohort 2) were still susceptible for severe fungal infection. This complication should be considered when patients are admitted to the ICU more than 6 months after successful HDT [24]. Including this additional cohort to the analysis elucidates further treatment necessities and their hitherto unknown complications, allowing clinicians to define goals and to predict outcomes for patients that were treated with high dose chemotherapy.

In our cohort, cut-off values for time of aplasia and Charlson comorbidity index were able to significantly predict ICU admission in a multivariate analysis. However, these predictors, although in concordance with clinical understanding, might not be able to influence clinical decision making in time to prevent ICU admission. In the future, more reliable markers may be helpful to predict the risk of ICU admission, thereby allowing for less urgent referrals of severely sick patients and leading to a better outcome [25,26].

Not only do our data strengthen the recommendation of the consensus statement to use a full treatment code but they also add independent risk factors for the admission to ICU after high dose chemotherapy and ASCT, such as age, duration of aplasia, and Charlson comorbidity index, as guidance parameters. However, the number of admitted patients to the ICU in this study as well as the retrospective character might be limiting the prediction of other risk factors. Moreover, the heterogenicity of the treatment regimens and the cancer entities treated in a single referral centre could conceal cancer entity specific characteristics, thus leading to an ICU survival.

## 4. Methods

This study is based on a retrospective analysis of prospectively collected data. All patients who received high dose chemotherapy with subsequent autologous stem cell transplantation in a tertiary medical centre from 1 January 2008 to 31 December 2014 were eligible for inclusion in this study. Patient data were censored at the time of data cut-off, which was performed on 31 July 2018.

Data were collected through electronical patient data management system (PDMS, Integrated Care Manager^®^ (ICM), Version 9.1—Draeger Medical, Luebeck, Germany). The extracted data included age, gender, comorbidities, underlying hematologic malignancy and course of disease, admission diagnosis, length of ICU/hospital-stay, treatment modalities and organ support (mechanical ventilation, vasopressor, renal replacement therapy, blood transfusions, antibiotics, antiviral treatment, etc.), laboratory parameters, and discharge information. Pre-existing medication was recorded on the basis of known regular medications and medication on admission. Survival follow-up data were acquired using the local cancer registry. Routine laboratory assessment was performed on daily basis within usual practice.

### 4.1. Study Definitions

Malignancies were defined and categorized as follows: refractory Hodgkin’s lymphoma (HL), non-Hodgkin’s lymphoma (NHL), primary central nervous lymphoma (CNS-L), sarcoma (S), germ cell tumour (GC), and multiple myeloma (MM). Furthermore, we defined disease status according to the guidelines published by the German society of haematology and medical oncology (DGHO). Patients admitted to the ICU were grouped in two cohorts. Cohort 1 included patients admitted in relation to complication during their HDT treatment within 30 days after ASCT. Cohort 2 was patients admitted to the ICU within the first 6 months after HDT excluding patients from cohort 1. Neutropenia was defined as a neutrophil count less than 0.5 × 10^9^ cells/L. This study was approved by the ethics committee of the Hamburg chamber of physicians (WF-111/20). Informed consent was waived due to the observational character of this study.

### 4.2. Statistical Analysis

Descriptive statistics were used to summarize the data. Results are presented as count and relative frequency or median and 25–75% interquartile ranges (IQR), as appropriate. Odds ratio was calculated using a multivariate binary regression model. Receiver operating characteristic (ROC) and area under the curve (AUC) analyses were used to assess the prognostic capacity of days of aplasia and Charlson comorbidity sum (Appendix A). Cut-off values were calculated via the Youden’s Index. Normal distribution was assessed using a D’Agostino’s K-squared test. Not normally distributed data were analysed using a Mann–Whitney-U-Test. Statistical analysis was conducted using IBM SPSS Statistics Version 24.0 (IBM Corp., Armonk, NY, USA) and GraphPad Prism (Version 6). A *p*-value of < 0.05 was considered statistically significant.

## 5. Conclusions

In conclusion, treatment regimens using HDT and subsequent ASCT in a specialised referral centre are safe. ICU-treatment is warranted mainly due to infectious complications and has a positive impact on intermediate-term survival and should therefore always be promoted.

## Figures and Tables

**Figure 1 cancers-12-01678-f001:**
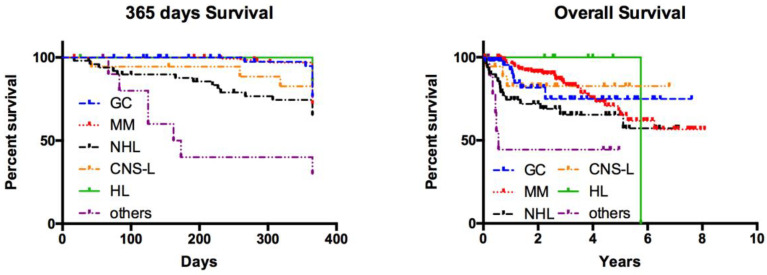
The 365-day survival and overall survival of patients treated with high dose chemotherapy and subsequent ASCT. Median survival over the period of follow-up was reached in the sarcoma and rare tumour group (median overall survival was 167 days).

**Table 1 cancers-12-01678-t001:** Demographic data and cancer diagnosis of patients receiving high dose chemotherapy and subsequent autologous stem cell transplantation (ASCT). Cohort 1: patients admitted to the intensive care unit (ICU) due to immediate complications during high dose chemotherapy (HDT) and ASCT treatment. Cohort 2 patient admitted within 6 months after high dose chemotherapy and ASCT (excluding cohort 1 patients).

Variable	All Patient	ICU Admitted Cohort 1	ICU Admitted Cohort 2
*n* (%) or Median [IQR]	(*n* = 256)	(*n* = 15)	(*n* = 31)
Demographic Data			
Age HDT	56 (45–4)	63 (43–67)	43 (27–50)
Age ≥ 60 Years at HDT	104 (41)	9 (60)	3 (10)
Female:Male	84 (33):172 (67)	7 (47):8 (53)	3 (10):28 (90)
Tumour			
Germ Cell Tumour	48 (19)	2 (13)	16 (52)
Seminoma	6 (2)	0 (0)	0 (0)
Non-Seminoma	37 (14)	1 (7)	16 (51)
Mixed	5 (2)	1 (7)	0 (0)
Multiple Myeloma	122 (48)	3 (20)	5 (16)
Non-Hodgkin-Lymphoma	49 (19)	5 (33)	6 (19)
T-cell-NHL	9 (4)	1 (7)	2 (6)
B-cell-NHL	40 (16)	4 (27)	4 (13)
DLBCL	20 (8)	0 (0)	3 (10)
Follicular-L.	6 (2)	1 (7)	0 (0)
Mantle cell-L.	7 (3)	1 (7)	0 (0)
Others	7 (3)	2 (13)	1 (3)
CNS-Lymphoma	18 (7)	2 (13)	1 (3)
Hodgkin-Lymphoma	8 (3)	0 (0)	1 (3)
Miscellaneous	11 (4)	3 (20)	2 (6)

IQR: interquartile range; ICU: intensive care unit; HDT: high dose chemotheraphy; NHL: non-Hodgkin’s lymphoma; CNS: central nervous system; DLBCL: diffuse large B-cell lymphoma.

**Table 2 cancers-12-01678-t002:** Demographic data and clinical parameters of patient. Cohort 1: patients admitted to the ICU due to immediate complications during HDT and ASCT treatment. Cohort 2 patient admitted within 6 months after high dose chemotherapy and ASCT (excluding cohort 1 patients).

Variable	ICU Admitted Cohort 1	ICU Admitted Cohort 2	*p*-Value
*n* (%) or Median [IQR]	(*n* = 15)	(*n* = 31)	
Demographic Data			
Age at ICU Admittance	64 (43–67)	44 (28–50)	
Age ≥ 60 at ICU Admittance	9 (60)	4 (13)	
Female:Male	7 (47):8 (53)	3 (10):28 (90)	
Time (days) between HDT and ICU	14 (11–15)	93 (56–122)	
Treatment Related Complication			
Aplasia (≤ 0, 5 × 10^9^/L) Admittance (ICU)	12 (80)	1 (3)	
Duration of Aplasia (Days)	7 (6–8)	3 (4–7)	
SOFA	8 (6–12)	4 (2–8)	0.001
Charlson Comorbidity-Index	4 (4–6)	4 (2–6)	
Cardiac Output (Ejection Fraction (EF))			
Normal (EF > 55)	7 (47)	14 (45)	
Reduced	3 (20)	3 (10)	
Not Defined	5 (33)	14 (45)	
Diagnosis at ICU Admittance			
Sepsis/Septic Shock	12 (80)	4 (13)	
With Respiratory Failure	7 (47)	3 (6)	
Cardiac Event (CPR)	1 (7)	0 (0)	
Neurological Disability	1 (7)	2 (6)	
Bleeding	0 (0)	3 (10)	
Ileus	1 (7)	0 (0)	
Post Elective Surgery	0 (0)	19 (61)	
Supportive Therapy			
Vasopressor	10 (67)	11 (35)	
Invasive Ventilation	8 (53)	2 (6)	
Dialysis	5 (33)	2 (6)	
Time on ICU (days)	8 (2–27)	1 (1–5)	0.0004
Time in Hospital (days)	46 (24–75)	14 (10–27)	<0.0001
Outcome			
Death on ICU	3 (20)	1 (3)	
356 Days Mortality after ICU Discharge	6 (50)	8 (27)	
Cause of Death			
Tumour Progression within 365 Days	4 (33)	5 (17)	

**Table 3 cancers-12-01678-t003:** Multivariant binary regression analysis of independent risk factors for ICU admission in relation to their HDT and ASCT treatment. **A:** continuous variable. **B:** binary variable.

**(A)**			
**Variable**	**Regression Coefficient**	**OR (95% CI)**	***p*-Value**
Duration of Aplasia	0.312	1.37 (1.07–1.75)	0.013
Charlson Score Sum	0.492	1.64 (1.20–2.30)	0.002
Age at HDT	0.048	1.05 (1.00–1.09)	0.043
**(B)**			
**Variable**	**Regression Coefficient**	**OR (95% CI)**	***p*-Value**
Aplasia (>6.5 days)	1.58	4.84 (1.61–14.45)	0.005
Charlson Score (>3.5)	1.73	5.65 (1.71–18.62)	0.004

**Table 4 cancers-12-01678-t004:** Reason for admission and infection focus in patients admitted to the ICU. Cohort 1: patients admitted to the ICU due to immediate complications during HDT and ASCT treatment. Cohort 2 patient admitted within 6 months after high dose chemotherapy and ASCT (excluding cohort 1 patients). HAP: hospital acquired pneumonia; VT: ventricular tachycardia; CPR: cardiopulmonary resuscitation; EVD: extraventricular drainage.

Variable	ICU Admitted Cohort 1	ICU Admitted Cohort 2
*n* (%)	(*n* = 15)	(*n* = 31)
Infection	12 (80)	8 (26)
Type of Infection		
Pneumonia (HAP)	8 (53)	6 (19)
Catheter Infection	1 (7)	0 (0)
Unknown Focus	3 (20)	2 (6)
Sepsis	10 (67)	3 (10)
Septic Shock	2 (13)	1 (3)
Bacteraemia	0 (0)	1 (3)
Non Infection Related Diagnosis	3 (20)	23 (74)
VT including CPR	1	0
Seizure	1	1
Ileus	1	1
Bleeding	0	2
Post Surgery	0	19 (61)
Tumour Resection Germ Cell Tumour (Thoracic)	0	11
Tumour Resection Germ Cell Tumour (Abdominal)	0	4
Implantation of EVD for NHL Brain Metastasis	0	1
Cerebral Biopsy NHL Metastasis	0	1
Osteosynthesis for Osteolysis at Multiple Myeloma	0	2

**Table 5 cancers-12-01678-t005:** Microbiological analysis of infections in patients admitted to the ICU. Cohort 1: patients admitted to the ICU due to immediate complications during HDT and ASCT treatment. Cohort 2 patient admitted within 6 months after high dose chemotherapy and ASCT (excluding cohort 1 patients) VRE: vancomycin resistant enterococcus; ESBL: extended spectrum β-lactamase producing bacteria. BAL: bronchoalveolar lavage

Variable	ICU Admitted Cohort 1	ICU Admitted Cohort 2
	(*n* = 15)	(*n* = 31)
Gram Positive Bacteria	17	6
BAL	8	5
Coagulase Negative Staph.	1	0
*Staphylococcus haemolyticus*	1	0
*Streptococcus Viridans*	3	0
Enterococcus Species	3	2
Corynebacterium	1	1
Blood Culture	9	3
Coagulase Negative Staph.	7	2
*Staphylococcus Epidermidis*	5	0
*Staphylococcus Haemolyticus*	2	1
*Staphylococcus Hominis*	0	1
Viridans Streptokokken	1	0
Enterococcus Species	1	1
Gram Negative Bacteria	4	1
BAL	2	0
*Escherichia Coli*	1	0
*Pseudomonas Aeruginosa*	1	0
Blood Culture	2	1
*Escherichia Coli*	1	1
HSV Pneumonia	1	0
Severe Fungal Infection	5	4
Candidasis	2	1
BAL	1	1
Blood Culture	1	0
Aspergillosis in BAL	3	3
Bacterial Resistance (Detected in Hospital Stay)		
VRE	4	1
ESBL	1	1
Carbapenemase (OXA48)	1	0

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
