# Peer review of "Intensive Care Outcomes of Patients after High Dose Chemotherapy and Subsequent Autologous Stem Cell Transplantation: A Retrospective, Single Centre Analysis"

_cancers, 2020, doi:10.3390/cancers12061678_

Round 1
Reviewer 1 Report
The manuscript reports the results of a retrospective analysis including 256 patients with different cancer entities treated at a single Center with high dose chemotherapy and autologous stem cell transplantation. Forty-six of these patients admitted to ICU for several complications either early or late (<6 months) after transplant were studied for outcomes. The Authors conclude that ICU-treatment has a strong positive impact on intermediate-long term survival.
Although the manuscript is well written and the analysis is accurate, the study lends itself to some criticism:
The heterogeneity of the patient series in terms of underlying disease, the different pre-transplant therapies in terms of intensity and drug combinations, the low number of patients and the retrospective nature of the analysis are the main factors which significantly limit the impact of this study.
Furthermore, what is the relevance of comparing patients admitted early with patients admitted late to ICU?
In order to prevent such event and to inform more properly patients and families, it might be more interesting to explore pre-transplant factors predicting the risk to be admitted to ICU.
Is overall survival different between patients not admitted and those admitted early to ICU?
The higher number of patients is represented by subjects suffering of myeloma or non Hodgkin lymphoma, so the analysis could be restricted to these two categories of patients. Considering that these 2 patient groups likely received each an identical pre-transplant conditioning regimen, a comparison between them could be interesting to investigate according with the spirit of this analysis.
The Authors analyze patients late (<6 months) admitted to ICU by excluding patients admitted early: why? Which is the relevance to compare these heterogeneous patient population by early and late ICU admission ? One question is: does an early ICU admission affect the occurrence and outcome of the second one ?
The analysis of transplanted patients late admitted to ICU including patients early ICU hospitalized could be matter for another study.
Finally, the Authors are invited to control the number reported in the tables with greater accuracy: e.g. in Table 1, the number of patients by tumour is 393 (256) in the first column, 25 (15) in the second and 57 (31) in the third one.
Author Response
Specific comment #1: The heterogenicity of the patient series in terms of underlying disease, the different pre-transplant therapies in terms of intensity and drug combinations, the low number of patients and the retrospective nature of the analysis are the main factors which significantly limit the impact of this study.
We do agree with the reviewer that the retrospective character of the study is a limitation. However, including different cancer entities could be a hitherto advantage as it does not only include the data from underrepresented cancers such as germ cell tumours but might also allow to overcome bias due to cancer entity specific characteristics. Considering the possible heterogenicity of pre-transplant therapies, our study can investigate the outcome of patients undergoing high-dose chemotherapy in general as well as elucidating that the type of high-dose therapy protocol is not representing a significant risk factor for adverse events, with the exception of protocols involving total body irradiation. The latter being not included in this study. However, we are convinced that the high number of patients analysed with 256 patients who received high dose chemotherapy and subsequent ASCT, this study might represent one of the largest analysis in terms of intensive care outcomes.
To highlight the reviewer’s valid argument, we have added more information to the discussion. Line 184-185 page 7
Specific comment #2: Furthermore, what is the relevance of comparing patients admitted early with patients admitted late to ICU?
Being able to compare patients who were admitted in relation to their treatment and compare them to patients post treatment allows to identify possible risk factors or predictors for intermediate term outcome. For example, assuming that cohort 1 is still immunosuppressed while cohort 2 is not, leads to treatment differences in certain differential diagnostic settings. Specifically, we were able to show that patients being admitted to the ICU within 6 months (cohort 2) are still susceptible to severe fungal infections. Therefore, investigation of cohort 2 allows to delineate whether high dose chemotherapy leads to prolonged risk such as infectious complications. Furthermore, by adding a cohort of oncological patients that have been treated with chemotherapy and require admission to ICU allows to predict long-term outcome of post therapy patients giving clinical insights and allowing to define clinical admission criteria.
Specific comment 3: In order to prevent such event and to inform more properly patients and families, it might be more interesting to explore pre-transplant factors predicting the risk to be admitted to ICU.
We thank the review for this specific comment and totally agree that pre-transplant parameters are important for informed consent of patient and to inform their families. We did asses pre-transplant parameters of which we reported the statistically significant predictors in table 3. We now added also the non-statically significant parameters which were evaluated in the results section: Line 100-101 page 4
Specific comment 4: Is overall surivial different between patients not admitted and those admitted early to the ICU?
Indeed, overall survival of cohort 1, cohort 2 and all non-ICU admitted patient is different. We are providing a graph for illustration for the reviewer’s discretion. As we are able to show intermediate term outcome for cohort 1 is still considerably significant with a median overall survival of 2.13 years. Furthermore, even if median survival for cohort 2 is not reached it is important to notice that even late admittance to ICU effects the immediate survival compared to non-ICU admitted patients
Specific comment 5: The higher number of patients is represented by subjects suffering of myeloma or non-Hodgkin lymphoma, so the analysis could be restricted to these two categories of patients. Considering that these 2 patient groups likely received each an identical pretransplant conditioning regime, a comparison between them could be interesting to investigate according with the spirit of this analysis.
We have provided a table for the discretion of the reviewer elucidating the chemotherapy therapies before high dose chemotherapy for multiple myeloma and lymphoma including their outcome (see below). Although a high number of patients received the same therapy there is still a diversity in treatment even in these two groups. As mentioned above we think that the heterogenicity of pre-transplant therapies can investigate the outcome of patients undergoing high-dose chemotherapy in general and therefore adds to the validity of the study by providing long-term outcome for a more representative population. We did however address the reviewer’s concern by mentioned it in lines 191-192 page 7.
Induction therapy of patients with multiple myeloma and lymphoma.
|
Variable n |
Patients (n=189) |
|||||||
|
Tumour |
Induction therapy |
Response after induction therapy |
||||||
|
|
|
MR |
PR/ VGPR |
CR |
SD/PD |
NA |
||
|
Multiples myeloma
B-cell-NHL
T-cell-NHL
CNS lymphoma
|
122
40
9
18
|
VCD VD RAD AD RD others
R-DHAP R-CHOP R-ICE others
DHAP CHOP ICE others
IELSG regime Freiburg regime |
75 31 3 3 5 5
19 12 17 4
3 4 1 1
10 8
|
3 0 0 0 0 1
2 2 2 0
0 0 0 0
0 0
|
61 26 3 1 3 1
10 5 7 3
0 4 0 0
8 6
|
7 1 0 1 0 0
3 4 2 0
3 0 0 0
1 0
|
1 3 0 0 2 0
4 1 4 0
0 0 1 1
1 0
|
3 1 0 1 0 3
0 0 0 1
0 0 0 0
0 2
|
VCD, Bortezomib, Cyclophosphamid, Dexamethason; VD, Bortezomib, Dexamethason; RAD, Lenalidomid, Adriamycin, Dexamethason; AD, Adriamycin, Dexamethason; RD, Lenalidomid, Dexamethason; R-DHAP, Rituximab- Dexamethason, Cytarabin, Cisplatin; R-CHOP, Rituximab- Cyclophosphamid, Hydroxydaunorubicin, Vincristin, Predniso(lo)n; R-ICE, Rituximab- Ifosfamid, Carboplatin, Etoposid; BEACOPP, Cyclophosphamid, Etoposid, Adriamycin, Procarbazin, Vincristin, Bleomycin, Predniso(lo)n; IELSG regime and Freiburg regime: Are study protocols including high dose Methotrexate and Cytarabine/ Thiotepa with or without Rituximab and followed by high dose BCNU/ Thiotepa with ASCT.
Specific comment 6: The authors analyse patients late (<6months) admitted to ICU by excluding patients admitted early: why? Which is the relevance to compare these heterogenous patient population by early and late ICU admission? One question is: does an early ICU admission affect the occurrence and outcome of the second one?
As mentioned above in specific comment 2 by adding a cohort of oncological patients that have been treated with chemotherapy and require admission to ICU allows to predict long-term outcome of post therapy patients giving clinical insights and allowing to define clinical admission criteria. Investigation of cohort 2 allows to delineate whether high dose chemotherapy leads to prolonged risk such as infectious complications. It is very difficult to predict if an early admission would affect the occurrence and outcome of a late admission. However, in our study only one patient that was admitted in cohort 1 was re-admitted in cohort 2 indicating that early admission is most likely not associated with late admission.
Specific comment 7: The analysis of transplanted patients late admitted to ICU including patients early hospitalized could be matter for another study.
As mentioned above analysing a late cohort would allow to delineate whether high dose chemotherapy leads to prolonged risk for infectious complications as well as other currently non-defined medical conditions. We highly agree with the reviewer that a sperate analysis addressing this question is very important.
Specific comment 8: Finally, the authors are invited to control the number reported in the tables with greater accuracy: e.g. in table 1, the number of patients by tumour is 393 (256) in the first column, 25 (15) in the second and 57 (31) in the third one.
We have improved the format of the table to make the reading easier. Please excuse any confusion of numbers that might being caused by the presentation of the table. We have added an edited example of table 1 to this specific comment to present the improvement.
|
|
||||||
|
Variable |
All patient |
ICU admitted cohort 1 |
ICU admitted cohort 2 |
|||
|
N (%) or Median [IQR] |
(N=256) |
(N=15) |
(N=31) |
|||
|
Demographic data |
|
|
|
|||
|
|
Age HDT |
56 [45;64] |
63 [43;67] |
43 [27;50] |
||
|
|
Age ≥60 years at HDT |
104 (41) |
9 (60) |
3 (10) |
||
|
|
Female: Male |
84 (33):172(67) |
7 (47):8(53) |
3 (10):28(90) |
||
|
Tumour |
|
|
|
|||
|
|
Germ cell tumour |
48 (19) |
2 (13) |
16 (52) |
||
|
|
Seminoma |
6 (2) |
0 (0) |
0 (0) |
||
|
|
Non-seminoma |
37 (14) |
1 (7) |
16 (51) |
||
|
|
Mixed |
5 (2) |
1 (7) |
0 (0) |
||
|
|
Multiple Myeloma |
122 (48) |
3 (20) |
5 (16) |
||
|
|
Non-Hodgkin-Lymphoma |
49 (19) |
5 (33) |
6 (19) |
||
|
|
T-cell-NHL |
9 (4) |
1 (7) |
2 (6) |
||
|
|
B-cell-NHL |
40 (16) |
4 (27) |
4 (13) |
||
|
|
DLBCL |
20 (8) |
0 (0) |
3 (10) |
||
|
|
Follicular- L. |
6 (2) |
1 (7) |
0 (0) |
||
|
|
Mantle cell-L. |
7 (3) |
1 (7) |
0 (0) |
||
|
|
Others |
7 (3) |
2 (13) |
1 (3) |
||
|
|
CNS-Lymphoma |
18 (7) |
2 (13) |
1 (3) |
||
|
|
Hodgkin-Lymphoma |
8 (3) |
0 (0) |
1 (3) |
||
|
|
Misellaneous |
11 (4) |
3 (20) |
2 (6)
|
||
Reviewer 2 Report
This is a well-written manuscript and the data are clearly presented. The use of autologous stem cell transplant (ASCT) for malignant diseases is increased nowadays. The results of this manuscript can add value to the literature on the outcomes and survival after intensive care unit (ICU) care for patients undergoing ASCT for malignant diseases.
There are some suggestions for the authors.
- There are two cohorts of patients in the study. Can the authors elaborate more on the rationale of studying the patients in cohort 2 (admission to ICU with first 6 months after ASCT)? Patients after high-dose therapy usually have neutropenia in the first four weeks and they are more prone to infection in this period.
- About half of the studied patients suffer from multiple myeloma (MM). The 5-year overall survival rates of MM patients of revised International Staging System (R-ISS) stage I, II and III are 82%, 62% and 40% respectively (Palumbo A et al. J Clin Oncol 2015; 33: 2863-9). The survival curves in the figure 1 of the manuscript showed that the 5-year overall survival rate of patients with MM was approximately 80%. Authors may discuss this point and the results of the present study can add evidence that ICU care may help to improve overall survival of patients with MM.
- There is a minor spelling error, “imidate complications” at line 133 of page 5 would be “immediate complications”
Author Response
Specific comment 1: There are two cohorts of patients in the study. Can the authors elaborate more on the rationale of studying the patients on cohort 2 (admission to ICU with first 6 months after ASCT)? Patients after high-dose therapy usually have neutropenia in the first four weeks and they are more prone to infections in this period.
Being able to compare patients who were admitted in relation to their treatment and compare them to patients post treatment allows to identify possible risk factors or predictors for intermediate term outcome. For example, assuming that cohort 1 is still immunosuppressed while cohort 2 is not, leads to treatment differences in certain differential diagnostic settings. Specifically, we were able to show that patients being admitted to the ICU within 6 months (cohort 2) are still susceptible to severe fungal infections. Therefore, investigation of cohort 2 allows to delineate whether high dose chemotherapy leads to prolonged risk such as infectious complications. Furthermore, by adding a cohort of oncological patients that have been treated with chemotherapy and require admission to ICU allows to predict long-term outcome of post therapy patients giving clinical insights and allowing to define clinical admission criteria.
Specific comment 2: About half of the studied patients suffer from multiple myelopma (MM). The 5-year overall survival rates of MM patients of revised International Staging System (R-ISS) stage I, II and III are 82%, 62% and 40 % respectively (Palumbo A et al J Clin Oncol 2015; 33:2863-9) The survival curves in the figure 1 of the manuscript showed that the 5 year overall survival rate of patients with MM was approximately 80%. Authors may discuss this point and results of the present study can add evidence that ICU care may help to improve overall survival of patients with MM.
We thank the reviewer for pointing out that the survival curves in figure 1 are not completely clear. We have now improved the figure for overall survival (see below). As pointed out by the reviewer the survival is around 60-70%. We also have added the mentioned publication to the main text of the manuscript lines 147-148 page 6.
Specific comment 3: There is a minor spelling error, “imidate complications” at line 133 of page 5 would be immediate complications.
Please excuse the mistake. This typographic error has been corrected and an additional round of proof reading was performed before resubmission.
Round 2
Reviewer 1 Report
Major revisions favourably acquired.